# Research Advances in Heterotrimeric G-Protein α Subunits and Uncanonical G-Protein Coupled Receptors in Plants

**DOI:** 10.3390/ijms22168678

**Published:** 2021-08-12

**Authors:** Ying Liu, Xiaoyun Wang, Danhui Dong, Luqin Guo, Xiaonan Dong, Jing Leng, Bing Zhao, Yang-Dong Guo, Na Zhang

**Affiliations:** Department of Vegetables, College of Horticulture, China Agricultural University, Beijing 100193, China; niudaying@cau.edu.cn (Y.L.); wangxiaoyun@cau.edu.cn (X.W.); S20193172461@cau.edu.cn (D.D.); luqin2019@cau.edu.cn (L.G.); s20203172666@cau.edu.cn (X.D.); SY20203172740@cau.edu.cn (J.L.); Zhaobing@cau.edu.cn (B.Z.)

**Keywords:** Gα, signal cycle, self-activation, uncanonical GPCR

## Abstract

As crucial signal transducers, G-proteins and G-protein-coupled receptors (GPCRs) have attracted increasing attention in the field of signal transduction. Research on G-proteins and GPCRs has mainly focused on animals, while research on plants is relatively rare. The mode of action of G-proteins is quite different from that in animals. The G-protein α (Gα) subunit is the most essential member of the G-protein signal cycle in animals and plants. The G-protein is activated when Gα releases GDP and binds to GTP, and the relationships with the GPCR and the downstream signal are also achieved by Gα coupling. It is important to study the role of Gα in the signaling pathway to explore the regulatory mechanism of G-proteins. The existence of a self-activated Gα in plants makes it unnecessary for the canonical GPCR to activate the G-protein by exchanging GDP with GTP. However, putative GPCRs have been found and proven to play important roles in G-protein signal transduction. The unique mode of action of G-proteins and the function of putative GPCRs in plants suggest that the same definition used in animal research cannot be used to study uncanonical GPCRs in plants. This review focuses on the different functions of the Gα and the mode of action between plants and animals as well as the functions of the uncanonical GPCR. This review employs a new perspective to define uncanonical GPCRs in plants and emphasizes the role of uncanonical GPCRs and Gα subunits in plant stress resistance and agricultural production.

## 1. Introduction

In 1994, the Nobel Prize in Medicine was awarded to Drs. Gilman and Rodbell for their breakthrough in purifying the G-protein. In 2012, the Nobel Prize in Chemistry was awarded to Drs. Lefkowitz and Kobika, who isolated the β-adrenoceptor as the first G-protein-coupled receptor (GPCR). The ability to perceive the environment and respond appropriately is a key factor for survival, and the G-protein signaling cascade is one of the main sensing mechanisms used by multicellular organisms.

Heterotrimeric G-proteins are ubiquitous signaling macromolecules in eukaryotes that regulate transmembrane signaling by coupling to the GPCR located on the cell surface. Whole-genome sequencing has shown that the heterotrimeric G-protein signal is highly complex. The human proteome contains 23 Gα, 5 Gβ, and 12 Gγ subunits [1], which could theoretically form more than 1300 heterotrimeric complexes. In sharp contrast to the abundant number and combination of G-proteins in animals, the number of heterotrimeric signaling components in plants is much less; additionally, it is interesting that plants have atypical G-proteins. The *Arabidopsis* genome encodes four Gα (GPA1, XLG1, XLG2, and XLG3), one Gβ (AGB1), and three Gγ (AGG1, AGG2, and AGG3) subunits [2]. Among them, GPA1 is a typical Gα subunit, similar to those in animals, while XLG1, XLG2, and XLG3 are atypical Gα proteins. In addition to the C-terminal Gα domain, these atypical Gα proteins contain an N-terminal domain with unknown function [3]. Compared with the typical Gα, atypical Gα has been less researched, and its role in the signaling pathway is not as important as the typical Gα. Therefore, this review will focus on the typical Gα in the following description.

The difference between the typical Gα in plants and the Gα in animals is that the Gα in plants can exchange GTP/GDP spontaneously [4], so it was previously thought that canonical GPCRs did not exist in plants. Signal transmission is a process of acceptance–transmission–response. The self-activation of Gα in plants must be stimulated by upstream signals. Therefore, for a long time, scientists have been committed to finding putative GPCRs in plants. Some transmembrane proteins have been proved to be upstream of Gα, and their signal transduction is completed through G-proteins. This proved that there should be putative GPCRs in plants to complete the G-protein signal cycle. This review will discuss the role of Gα in plants and propose a new definition of plant uncanonical GPCRs. By combining known pathways and new canonical GPCR definitions, we propose a new plant G-protein cycle model.

## 2. Self-Activation and Recycling of Gα in Plants

In animals, there are two domains in Gα; one is the Ras domain, which can bind to GTP, and the other is a helix domain. The Ras domain contains sites that can bind to GPCRs, regulators of the G-protein signaling (RGS) protein, GTP, and other substances, and the role of the helix structure is to surround the chimeric sites of guanine nucleotides to create a binding environment [5]. The Gα and Gβγ dimers combine to form an inactive trimer in the resting state, and Gα is bound to GDP. Under stimulation by extracellular ligands, GPCR, as a guanine nucleotide exchange factor (GEF), stimulates the guanine nucleotide-binding site of Gα to exchange GDP for GTP. GTP binding terminates the combination between the Gα and Gβγ dimers, resulting in dissociation and formation of activated GTP-Gα and Gβγ dimers. Both continue to regulate different downstream targets known as “effectors” to transmit signals. At the same time, GTP is hydrolyzed by the spontaneous GTPase activity of Gα, making Gα return to resting, bound to GDP, thus completing the signaling cycle of the heterotrimeric G-protein [5,6] (Figure 1). The RGS in the G-protein signaling cycle is the GTPase accelerating protein (GAP). Regulation of the GAP promotes the hydrolysis of GTP [7], and there are at least 37 kinds of RGS proteins in humans [8].

Typical Gα proteins have been found in all plants except *Physcomitrella patens* [9]. Unlike animals, *Arabidopsis* GPA1 exhibits a very high GTP/GDP exchange rate and a very slow GTP hydrolysis rate. In addition, the rate-limiting regulatory step in the plant G-protein cycle is inactivation, which is regulated by RGS proteins. Although *Arabidopsis* GPA1 also has a Ras functional domain and a helical structure, it has autoactivating activity and can complete the cycle in the absence of a GEF by binding to GTP to accomplish signaling while deactivation occurs through RGS proteins [4]. The unique local dynamics of helix α A, which is different from animal Gα, allows GPA1 to perform receptor-independent GDP release and GTP binding [10,11]. Gα combines with GTP, as in animals, and forms activated GTP-Gα and Gβγ dimers, which transduce the signal downstream. Under the action of other proteins such as GAP, GTP is hydrolyzed to restore Gα into a resting state that binds GDP (Figure 2). Therefore, the role of GAP (such as the RGS protein) is essential for regulating the G-protein cycle during the restoration of the resting state. The RGS in plants contains a functional domain with seven transmembrane domains (TMs) at the N-terminus, and the C-terminus contains an RGS region located in the cytoplasm. After RGS combines directly or indirectly with an exogenous stimulating ligand, C-terminal phosphorylation affects the hydrolysis of GTP [12]. A single amino acid substitution (Alanine 357 to Valine) of soybean RGS2 affected its interaction with Gα, which is responsible for a GAP activity change [13]. *Arabidopsis* has a single RGS1 [3] that negatively regulates GPA1-mediated signal transduction through its GAP activity. Therefore, the plant completes the G-protein signaling cycle by relying on the self-activation of Gα without stimulation by canonical GPCR and inactivation with RGS. The functional networks between the Gα and RGS are conserved in plants, and despite the absence of RGS in many monocots, their corresponding Gα retains the ability to be deactivated by non-native RGS in plants [14].

Four kinds of Gα proteins occur in soybean. Interestingly, when soybean Gαs were used to supplement the *Arabidopsis gpa1* mutant, the proteins GmGα2 and GmGα3 completely recovered each mutant phenotype, while the proteins GmGα1 and GmGα4 only supplemented part of them [15]. The soybean Gα protein, introduced into the yeast *gpa1* mutant, also showed differences. In yeast, GmGα1 and GmGα4 completely restored all growth and pheromone signal phenotypes of the yeast *gpa1* mutant, while GmGα2 and GmGα3 only partially supplemented these phenotypes [16]. Because yeast has the classical Gα activation of GDP/GTP exchange based on the canonical GPCR, at least part of the plant Gα protein can be activated by the canonical GPCR in a heterologous system, regardless of its own activation ability. In contrast, although it had the ability to self-activate, another group of Gα proteins did not fully function in the yeast system. The degree or rate of self-activation of Gα proteins in plants may change and affect their ability to regulate responses, indicating that there may be an alternative mechanism to promote Gα activation in plants, which may be another possible situation of G-protein cycle regulation [17].

By studying the constitutive activity of the GTP-bound AtGPA1 (Q222L) mutant and the nucleotide-free AtGPA1 (S52C) mutant, Maruta found that they could interact with Gγβ dimers and the GTP-binding-impaired AtGPA1 (S52C) variant complemented some, but not all, *gpa1*-null mutant phenotypes. This means that in addition to the classic GDP–GTP exchange-dependent mechanism, plant G-proteins may also have mechanisms that function independently of nucleotide exchange [18]. Therefore, this review will mainly focus on research in the canonical GDP–GTP exchange-dependent mechanism of G-proteins.

Recent research has reported that there are some pathways in plants that rely on the interactions between multi-transmembrane proteins and Gα to transmit signals, which will be described in detail below.

## 3. Research Progress on the Gα in Plants

The study of Gα is of great significance in animals and plants. In animals, it has primarily focused on the drug action pathway, while in plants, it pays attention to signal transduction, hormone responses, and biotic and abiotic stress resistance.

Growth and development at all stages of plants require regulation by hormones such as gibberellins (GAs), abscisic acid (ABA), and brassinosteroid (BR). The interactions between hormones and G-protein fine-tune many biological processes in plants; however, the molecular mechanisms are largely unknown. The functional characterization of hormone biosynthesis, perception, and signal components, as well as the identification of a small number of effector molecules of G-proteins and their interaction network, reduce the complexity of the hormone signal network related to G-proteins [19]. In this review, we summarize important progress in this field and show the latest work in *Arabidopsis* and rice, which is conducive to understanding the relationship between plant hormones and G-proteins.

Seeds of *Arabidopsis gpa1* mutants have a significantly longer dormancy period and exhibit insensitivity to exogenously applied GA [20], whereas the concentrations of endogenous ABA are similar in mutants and the wild-type. This finding indicates that the insensitivity of *gpa1* mutants to stimulatory signals such as GA and delays in germination [21]. Mutants of rice Gα *rga1* exhibited phenotypes such as dwarf and round grains [22]. The gene expression and enzyme activity of alpha-amylase induced by GA were significantly reduced, indicating that Gα was also involved in the GA pathway [23]. Epibrassinolase (EBR), a bioactive BR, induces stomatal closure in *Arabidopsis* leaves. EBR-induced stomatal closure was completely abolished in *gpa1-1* and *gpa1-2* mutants, demonstrating that Gα played an important role in EBR-induced stomatal closure. In pharmacological experiments with the Gα inhibitor PTX and the Gα activator CTX, the positive regulator Gα mediated EBR-induced stomatal closure in *Arabidopsis* [24]. In *Arabidopsis*, the *gpa1* mutant was insensitive to the inhibition of stomatal opening by ABA, and *gpa1* exhibited reduced inward potassium channel activity, which was inhibited by ABA [25]. Previous studies have shown that ABA and drought in *Arabidopsis* stimulate the production of the lipid metabolite sphingosine-1-phosphate (S1P) in plants, indicating that S1P is a second messenger in guard cell ABA responses and that the *gpa1* mutant is insensitive to the effects of S1P in regulating stomatal pores and ion channels [26]. These facts show that GPA1 plays an essential role in the ABA pathway. In the *Arabidopsis* seed germination test, BR enhanced GA-induced seed germination through heterotrimeric G-protein coupling, while ABA attenuated GA-induced seed germination; *gpa1* mutants were hypersensitive to ABA and GA, but insensitive to BR [27]. The plant hormone methyl jasmonate (MeJA) induces stomatal closure in many plant species, but it can be abolished in the guard cells of *Arabidopsis gpa1* mutants by disrupting MeJA-activated H^+^ efflux, Ca^2+^ influx, and K^+^ efflux. The accumulation of reactive oxygen species is also lower in *gpa1-1* and *gpa1-2* mutant guard cells under MeJA treatment than in the wild-type. These results suggest that the Gα subunit is involved in regulating the MeJA signaling pathway and signaling events during stomatal closure [28]. The rice Gα subunit, RGA1/D1, regulated ethylene (ET)- and H_2_O_2_-induced epidermal cell death; thus, *d1* mutant plants show strong inhibition of epidermal cell death in response to ET and H_2_O_2_ [29]. *Arabidopsis* GPA1 promotes stomatal closure by ethylene-induced NADPH oxidase-dependent H_2_O_2_ production. Ethylene activated GPA1 after binding to its receptors, and subsequent inactivation of CTR1 (constitutive triple response 1), in turn, induced H_2_O_2_ production in guard cells [30]. A certain concentration of auxin promotes cell division. *Arabidopsis* GPA1 plays a key role in auxin-mediated cell division. *Gpa1* mutants display reduced cell division in developing hypocotyls and leaves. In the process of lateral root development, GPA1-mediated cell division caused by auxin was inhibited by AGB1 [31,32] (Table 1). Phospholipase D (PLD) is the important regulator of plant signaling and metabolic pathways, especially G-protein-mediated hormone responses. PLDα1 has been linked to the G-protein cycle during the regulation of a subset of ABA-mediated responses [33]. In *Arabidopsis*, RGS1 and PLDα1 both act as GAPs for Gα to attenuate its activity. RGS1 and PLDα1 interact with each other, and RGS1 inhibits the activity of PLDα1 during the regulation of a subset of responses. This regulation is bidirectional. Phosphatidic acid (PA) typically derives from the lipid-hydrolyzing activity of PLDα1 and is a molecular target of RGS1. PA binds and inhibits the GAP activity of RGS1. Such developmental plasticity and interaction specificity likely compensate for the lack of multiplicity of individual subunits and help to fine-tune the plants’ responses to constantly changing environments [34,35]. GPA1 and PLDδ are involved in the regulation of JA under osmotic stress. Both GPA1 and PLDδ participate in the regulation of JA in seed germination and osmotic tolerance [36]. PLDα1 and GPA1 are involved in oridonin-induced stomatal closure, and PLDα1 acts downstream of GPA1. Oridonin caused stomatal closure by affecting GPA1 and promoting PLDα1 to produce PA and further accumulating H_2_O_2_ to upregulate the expression of gene *OST1* (open stomata 2/*Arabidopsis* H^+^-ATPase 2) [37].

Gα plays an important role in plant growth and development, particularly in the expansion and elongation of organs in commercially important crops. The hypocotyls of *Arabidopsis gpa1* and *agb1* mutants are shorter than those in the wild-type. There were more lateral roots in the *agb1* mutant compared with the wild-type but fewer in the *gpa1* mutant. Division of meristematic cells decreased during the entire developmental process, which might be related to the decrease in the axial division of epidermal cells in the hypocotyl, while the change in the number of lateral roots was caused by an activity change in the lateral root primordium [39]. The overexpression of *gpa1* in plants or cultured cells resulted in the division of ectopic cells and an accelerated cell cycle, which led to excessive cell division in the meristematic region and the initiation of an amorphous meristem, indicating that GPA1 is a positive regulator of plant cell division [31,40]. Further studies have reported that the GTP-binding form of GPA1 is a positive regulator, while the Gβγ dimer acts independently of Gα to weaken cell division. These results indicate that the G-protein subunits of the *Arabidopsis* heterotrimer have different or opposite roles regulating the division of root cells [41]. In a study of the rice *rga1* mutant, heterotrimeric G-protein signal transduction was related to the gibberellin response and resistance to pathogens. The *Rga1* mutants were dwarf, with smaller seeds, decreased α-amylase activity, and decreased GA-inducible aleurone gene expression [23,38]. The maize Gα mutant exhibited slower root development, clusters of ears, and thicker tassels than the wild-type, which might be related to inhibited growth and the development of lateral ears by Gα. Similarly, the maize Gα mutant *ct2* exhibited semi-dwarf height. These plants were about 32% shorter than the wild-type, and the erect leaves were about 31% shorter than the wild-type [42,43]. Plant roots avoid extracellular ATP as they bend and stay away from ATP-containing medium. The *Arabidopsis gpa1* mutant exhibited a weakened avoidance response but overexpression of *gpa1* responded strongly. This was related to a Ca^2+^ influx regulated by GPA1 and the asymmetric distribution of the PIN2 protein [44]. The secretion of organic acid anions (OAs) in plant roots occurs in response to aluminum, a common mechanism of plant resistance to aluminum. Aluminum ion-induced OA secretion was blocked in the *Arabidopsis gpa1* mutant, which was inhibited by the G-protein antagonist but stimulated by the cholera toxin. Moreover, aluminum ions also induced GPA1 expression in *Arabidopsis* roots, indicating that the Al stress signal transduced by GPA1 might be related to the secretion of OAs in roots. In processing tomatoes, LeGPA1 has a positive regulatory effect on cold-response gene expression. Overexpression of *LeGPA1* could alleviate cell membrane damage and accumulation of ROS under low-temperature stress, which enhanced the resistance of transgenic tomato seedlings to low temperature [45]. In oilseed *B. juncea*, the RNAi-based suppression of Gα genes resulted in multifarious effects on plant growth and development, such as reduced growth, smaller seeds, and less seed weight. Furthermore, over-expression of a Gα subunit enhanced plant height, organ size, and seed weight [46]. A major rice nitrogen-use efficiency quantitative trait locus, DEP1 (dense and erect panicles 1), interacted in vivo with RGA1, reduced RGA 1 activity, and inhibited nitrogen responses. This shows that the plant G-protein complex regulates nitrogen signaling and the modulation of heterotrimeric G-protein activity and thus provides a strategy for environmentally sustainable increases in rice grain yield [47]. In plant cells, the aquaporin (AQP) channel proteins facilitate the transport of water, primarily through the plasma and tonoplast membranes, and are designated as plasma membrane intrinsic proteins (PIPs) or tonoplast intrinsic proteins (TIPs), respectively. In cucumber, CsGPA1 interacted with CsTIP1.1, and the suppression of *CsGPA1* resulted in opposite patterns of expression of *CsAQPs* in leaves and roots, resulting in the declined water content of cucumber under salt stress [48].

In both animals and plants, Gα can bind receptor kinases (RKs), receptor-like kinases (RLKs), and receptor-like proteins (RLPs), which are single transmembrane proteins. Maize RLP FEA2 was shown to associate with the Gα protein CT2 to maintain the development of the shoot apical meristem [43], and soybean RK NFR1 interacted with Gα to control nodulation [49]. *Arabidopsis* RK ERECTA interacted with GPA1 to regulate disease resistance [50]. ERECTA is also a positive regulator of cantil formation, which is an unreported macroscopic *Arabidopsis* organ, named for its ‘cantilever’ function of holding the pedicel at a distance from the stem. ERECTA functions genetically upstream of heterotrimeric G-proteins. Cantil expressivity was inhibited by GPA1 [51]. In *Arabidopsis*, GPA1 responded to a bacterial flg22 elicitor and played a vital role in the immune pathway involving the flg22 receptor FLS2, co-receptor RLK BAK1, RGS1, and AGB1, in which flg22 promoted GPA1/AGB1 dissociation from the FLS2/BAK1/RGS1 receptor complex. In this way, BAK1 was likely the kinase for GPA1 phosphorylation in response to flg22 signaling [52,53].

G-protein signal transduction is the core of crop physiology research. The research on G-protein signal transduction will promote the development of agriculture.

## 4. GPCR, a G-Protein-Coupled Receptor

GPCRs are the largest family of membrane proteins. They mediate most cellular responses to hormones and neurotransmitters and are responsible for vision, olfaction, and taste. At the most basic level, canonical GPCRs are characterized by the presence of seven transmembrane α-helix segments separated by alternating intracellular and extracellular loops [54]. The known GPCR family binds to a wide range of ligands, including small organic compounds [55], eicosanoids [56], peptides [57], and proteins. The classical role of GPCRs is to couple the binding of ligands to the activation of specific heterotrimeric G-proteins, leading to the regulation of downstream effector proteins [54]. When GPCRs are activated by ligand binding, signals on the outside of the cell are initiated, which stimulate the exchange of GDP for GTP on the Gα subunit, with the subsequent dissociation of the heterotrimer into Gα subunits and Gβγ dimers. The Gα and βγ dimers continue to initiate the downstream signaling cascade [58].

### 4.1. Structure and Classification of Canonical GPCRs

As structure determines function, it is very important to understand the protein structure of the GPCRs. It was not until 1998 that researchers had their first understanding of the structure of GPCRs because most GPCRs are expressed at low levels in natural tissues and there are problems with thermodynamic and proteolytic stability. The first analytical breakthrough occurred with two-dimensional rhodopsin crystals. Rhodopsin is a transmembrane protein that transfers energy from light to an intracellular signaling cascade [59]. Through the development of radioligand binding and solubilization and the purification of monoamine-binding GPCR methods, scientists have made significant progress in the detection of the structure of GPCRs [60]. The structure of GPCRs can be divided into three sections: an extracellular domain, consisting of an N-terminal and three extracellular loops (ECL1–ECL3); a TM, consisting of seven alpha helices (TM1–TM7); and an intracellular domain, consisting of three intracellular loops (ICL1–ICL3), an intracellular hydrophilic–lipophilic helix (H8) and a C-terminus. The extracellular region generally regulates the access of ligands, and the TM forms the structural core, binds ligands, and transduces signals to the intracellular region, which is linked to intracellular signaling proteins through conformational changes [61].

Sequence analysis showed that the length and sequence composition of the N-terminal and extracellular loops were quite different, and the main functional structure was the extracellular loop. The two different types of extracellular domains are occluded ligand-binding pockets and hydrophilic ligand-binding pockets [62,63]. The special feature of the extracellular domain is the presence of disulfide bonds, which benefit receptor stability. The disulfide bonds of TM3-ECL2 anchor the extracellular region of the helix near the binding site and limit conformational changes in this region during receptor activation. The TM helix links the ligand-binding pocket and the G-protein coupling region during signal transduction. Although all GPCRs have a similar structure, which is connected by seven TM helices, their sequences are different. The residues in the cytoplasmic end and the intracellular region of the TM region bind to downstream signal effectors such as G-proteins, GPCR kinases, and repressors [64]. Activation of the GPCR involves the binding of the ligands to the extracellular part of the TM region, resulting in small conformational changes of the TM core protein. This results in the structural rearrangement of the transmembrane and intracellular regions. Therefore, the activation of the GPCR is defined as a receptor conformational change by coupling and stabilizing effector molecules (such as the heterotrimeric G-protein) [65].

### 4.2. Prediction of Putative GPCR in Plants

GPCRs are the largest family of transmembrane signal transduction proteins in multicellular organisms. Although GPCRs seem to exist only in eukaryotes, they are ubiquitous and have been cloned from many organisms with far-distant evolution, including yeasts [66], corals [67], nematodes [68], arthropods [69], humans [70], and preserved mammoth DNA [71]. Conservation of GPCR sequences may be less than 25% within a single GPCR family. Therefore, GPCRs cannot be identified by sequence homology but by their coupling ability with the intracellular heterotrimeric G-protein α subunit and their two-dimensional topology. Two-dimensional topology usually includes an extracellular amino-terminal, seven transmembrane domains connected by three intracellular rings and three extracellular rings, and a carboxyl-terminal tail [72].

Due to the low sequence similarity of GPCRs, it is necessary to identify new GPCRs using methods other than the Basic Local Alignment Search Tool (BLAST). Although previous studies believed that GPCRs did not exist in plants, there were still scientists looking for putative GPCRs in plants.

The QFC algorithm proposed by Kim et al. distinguishes GPCRs from non-GPCRs with a 98% success rate. The QFC algorithm includes four parameters, such as an amino acid usage index, a logarithm of the average hydrophobic function cycle, a logarithm of the average polarity scale cycle, and the variance of the first derivative of the polarity scale [73]. Based on the QFC algorithm, Gookin established the first criterion for identifying putative plant GPCRs in 2008. On the basis of the QFC-algorithm-predicted GPCR candidates, the prediction of signal peptides must be corrected first, and then, the 7-TM is predicted using three TM prediction programs (TMHMM2, HMMMTOP2, and Phobius). The GPCRHMM prediction method, based on amino acid composition and the topological fragment length between GPCR families, has also been employed. Gookin screened eight CANDs (candidate GPCRs) from 2469 *Arabidopsis* proteins meeting the QFC requirements using the method described above. Seven proteins interacted with GPA1 [74]. Although Gookin successfully predicted plant CANDs as putative GPCRs using the established method, the method still has some limitations. The quantitative requirement of 7-TM was based on the fact that almost all canonical GPCRs in animals have 7-TM, which resulted in many transmembrane proteins that did not have 7-TM but participated in a G-protein signaling pathway not being included as putative plant GPCRs.

## 5. Research Progress and New Definition of Uncanonical Plant GPCRs

In the study of the G-protein signal pathway, there are many transmembrane proteins involved in signal transduction upstream of the G-protein. They are the most likely putative plant GPCRs. The following will introduce these putative plant GPCRs in detail and propose a new definition of uncanonical plant GPCRs. This will help us find more uncanonical GPCRs in plants.

### 5.1. Putative Plant GPCRs

GCR1 in *Arabidopsis* is a membrane protein with 7-TM. It shows 20–23% homology with the GPCR family cAMP receptor (CARS) in *Dictyostelium discoideum* and conserved amino acid residues [75]. Pandey demonstrated the interaction between GCR1 and GPA1 using the split ubiquitin system, and the interaction required a free C-terminal in GCR1 [76]. GCR1 might be a negative regulator of the GPA1-mediated ABA response in guard cells. The lipid metabolite S1P is a signaling molecule in the ABA signaling transduction pathway that functions upstream of GPA1. Guard cells in wild-type plants are sensitive to S1P. The *gpa*1 mutant had no sensitivity to ABA or S1P in guard cells, while the *gcr1* mutant was hypersensitive. GCR1 also plays a positive role in the regulation of seed germination by GA and BR [21]. Chakraborty performed a transcriptomic analysis on the *Arabidopsis gpa1-5 gcr1-5* double mutant and identified 656 differentially expressed genes (DEGs). Further analysis of the DEGs revealed GCR1 and GPA1 could work together or independently to regulate different pathways [77,78].

GCR2 is another 7-TM membrane protein that has been demonstrated to interact with GPA1, and its C-terminal is necessary for its interaction with GPA1. The *gcr2* mutant was defective in all known ABA responses, while the overexpressing line was hypersensitive to ABA. The binding of GCR2 to ABA was specific and saturable and followed the rules of receptor kinetics, indicating that GCR2 was an ABA receptor. At the same time, the combination of ABA and GCR2 led to the dissociation of the GCR2-GPA1 complex, indicating that GCR2 and GPA1 might act together during ABA signal transduction [79].

Rice COLD1 is a 9-TM membrane protein that can interact with Gα on the cell surface. COLD1 contains nine transmembrane domains, with an extracellular N-terminal and intracellular C-terminal. *COLD1* encodes a protein located in the plasma membrane and endoplasmic reticulum (ER). COLD1 interacted with RGA1 (Gα) to trigger the Ca^2+^ channel and electric physiological signal changes once the cells were stimulated with cold temperature. Overexpression of *COLD1* significantly improved cold tolerance in rice, while *COLD1* deleted or downregulated lines that were sensitive to cold. A biochemical activity analysis confirmed that COLD1 alone did not have GTPase activity but promoted the GTPase activity of RGA1 [80]. In subsequent studies, through global analysis, researchers found that the vitamin E and K1 sub-network in chloroplast was downstream of COLD1; it is the key regulatory point for the formation of the difference in low-temperature tolerance between *indica* and *japonica* rice [81].

In 2018, researchers found the first plant melatonin receptor, CAND2, in *Arabidopsis*. CAND2 is a 7-TM protein that interacts with GPA1. *Arabidopsis* mutants *cand2* and *gpa1* are insensitive to melatonin for stomatal closure. Melatonin did not induce hydrogen peroxide production or internal Ca^2+^ flow in the mutant, while a G-protein inhibitor or activator weakened or enhanced the downstream signal. This finding indicated that melatonin-induced stomatal closure was achieved by the interaction between CAND2 and GPA1 and the regulation of hydrogen peroxide and Ca^2+^ signals in the process. The expression of *cand2* in various organs and guard cells is regulated by melatonin [82].

A novel putative GPCR named TOM1 (target of Myb1) has been reported; it enhances the tolerance of drought and cold stress by promoting root growth and the induction of ROS-scavenging enzymes [83,84]. However, the interaction mechanism of TOM1 and Gα needs to be further studied. Although there are so many putative plant GPCRs that have been found (Table 2), the above-mentioned putative plant GPCRs are not recognized as plant GPCRs as they have low or no sequence similarity with known canonical GPCRs in animals [85]. It is necessary to clearly define plant uncanonical GPCRs in order to study them better.

Significantly, previous studies have mainly focused on exploring whether putative plant GPCRs interact with Gα and their influences on downstream signals; how these interactions affect Gα from binding GDP to GTP and the structural requirements of the interactions were less studied.

Take COLD1, for example; once the cells are stimulated by cold temperature, COLD1 interacts with RGA1 to trigger the Ca^2+^ channel and electric physiological signal changes. In this process, that the interaction between COLD1 and RGA1 made RGA1 replace GDP with GTP is the first step. After completing the response to the cold signal, COLD1 accelerates the GTPase activity of RGA1 to produce a more GDP-bound state that might induce a regression shift on the equilibrium between GDP- and GTP-bound states of RGA1 for self-activation. Biochemical activity assays confirmed that RGA1, instead of COLD1 alone, had GTPase activity. RGA1 GTPase activity was accelerated in the presence of COLD1 [80]. The role of COLD1 in RGA1 exchanging GDP for GTP and hydrolyzing GTP might be due to the lack of typical RGS proteins in rice [14]. This allows COLD1 to play multiple roles in the G-protein cycle in response to cold signals. However, how other putative plant GPCRs affect the GDP–GTP exchange in Gα is unknown.

The structural requirements of interactions between plant putative GPCRs and Gα are worth studying. Previous studies in animal systems with chimeric and mutagenized GPCRs have shown ICL2 and, in particular, ICL3 to be the major determinants of coupling specificity between GPCRs and G-proteins [86]. In the split ubiquitin system, for GCR1, as in the mammalian systems, the presence of a complete ICL3 structure in GCR1 and/or some key amino acids from the beginning of ICL2 to the beginning of ICL3 are essential for its interaction with GPA1. The lack of structure of ICL3 makes the interaction undetectable in the split ubiquitin system. The second requirement for interaction is the presence of a free C-terminus in GCR1. GCR1 with C-terminal structure deletion and GCR1 with ubiquitin fused at its C-terminus failed to interact with GPA1 [76]. This result was different from the data of mammalian systems, in which the C-terminal cytoplasmic portion is not required for interaction [87]. There are relatively few studies on the structural requirements of interaction between other putative plant GPCRs with G-protein interactions. In future studies, mutagenesis of the C-terminal and a single residue in the intracellular ring of the putative GPCRs will help us to locate the exact amino acid sequence required for plant GPCR–GPA1 interaction.

### 5.2. New Definition of GPCRs and a Proposed Model of the G-Protein Cycle in Plants

In animal studies, GPCR is defined as a receptor enzyme with GEF activity. In other words, the canonical GPCR is a GEF receptor. This narrow definition originates from the finding that the activation of the Gα subunit in animals is caused by a GPCR stimulating the exchange site between GTP and GDP. However, as the Gα subunit in plants is self-activated, this narrow definition and identification method should not be applied to the search for plant uncanonical GPCRs. Although the plant G-protein is self-activated as a key component of a signaling pathway, there must be upstream and downstream signaling factors to complete signal transduction and the response. In the previous studies on G-proteins, researchers have always paid attention to the role of G-proteins and their downstream effectors, but there are few studies on why G-proteins are activated and what their upstream signal is. The parallel relationship between plant Gα self-activation and animal Gα exchange-activation strongly suggests that plant uncanonical GPCRs should have a unique definition.

The common features of putative plant GPCRs are that they are multi-transmembrane proteins, interact with Gα, and are upstream of the Gα in the G-protein signaling pathway. The ultimate role of GPCRs in animals is to activate the Gα subunit to transduce a signal. Therefore, we define the uncanonical GPCRs in plants as a kind of multi-transmembrane protein that receives extracellular ligand signals and interacts with Gα to transmit the signal. The uncanonical GPCR in plants does not exchange GTP and GDP, as in animals, but acts as a transmembrane signaling protein to transduce a signal to Gα for self-activation.

In plants, resting-state Gα carries GDP and binds to the Gβγ dimer. When the uncanonical GPCR is stimulated by an extracellular ligand, it interacts with Gα to transduce the signal. The Gα self-activates after receiving the signal, exchanges GDP for GTP, and enters the activation state of carrying GTP. The activated Gα and Gβγ dimers dissociate and bind to effectors to transduce the signal. After that, RGS protein hydrolyzes GTP to restore Gα to the resting state of binding to GDP, and the resting Gα and Gβγ recombine to complete the G-protein signal cycle (Figure 3). In the whole process, the hydrolysis of GTP by RGS is the key step.

Under this new definition, the previously reported COLD1, GCRs, and CAND2 can be classified as plant uncanonical GPCRs, as they play roles in the G-protein signaling pathway. As the new definition of plant uncanonical GPCRs is determined, the classic G-protein signaling cycle can also be fully explained in plants. The new definition helps us better understand the plant G-protein pathway and find more upstream and downstream signaling factors. This allows us to better study the important role of G-protein signals in plant growth, development, physiological, and biochemical processes.

## 6. Summary and Prospect

The G-protein and its GPCR are important components of signal transduction cascades. Studies on GPCRs in signal transduction, drug targets, and hormone receptors are increasingly being performed, and more and more studies on the mechanism of G-protein regulation of plant growth and development are being published. Although the G-protein plays an important role in regulating plant growth and stress resistance, the mechanisms of G-protein activation and inactivation and the downstream effector cycle remain unclear. Although coupling-activation of Gα occurs in animals, whereas self-activation of Gα occurs in plants, self-activation of Gα must be caused by stimulation from an upstream signal.

Here, we propose a new and unique definition of uncanonical GPCRs in plants. Uncanonical GPCRs in plants are defined as a class of multi-transmembrane proteins that perceive extracellular ligand signals and interact with the Gα subunit to transduce signals. Signal transduction is a complex process. It would be helpful to identify the upstream and downstream components of the G-protein signaling pathway to complete the G-protein pathway process.

Interestingly, the ABA and melatonin signals in response to GCR2 and CAND2 also affect the opening and closing of stomata through the heterotrimeric G-protein. The signaling pathways are coordinated and focus on the cellular responses that cause changes in guard cell volume, including activation/inactivation of ion channels, reorganization of the actin cytoskeleton, endocytosis/exocytosis, and changes in gene expression and regulation. The signaling molecules of these reactions include cytosolic calcium ions, protein kinases and phosphatases, reactive oxygen species, heterotrimeric G-proteins, and ROP small G-proteins [88]. Studies of these signaling molecules can be used to explore the G-protein signaling pathway through different perspectives.

Plant G-proteins and uncanonical GPCRs play an important role in stress resistance, growth, and development. The number and special types of putative plant GPCRs reported are also increasing. The mechanisms of G-protein activation or inactivation and the upstream and downstream signal transduction pathways are worthy of future study.

## Figures and Tables

**Figure 1 ijms-22-08678-f001:**
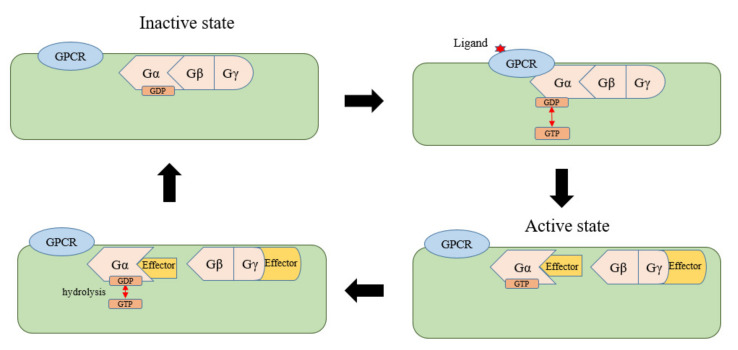
G-protein signaling cycle in animals.

**Figure 2 ijms-22-08678-f002:**
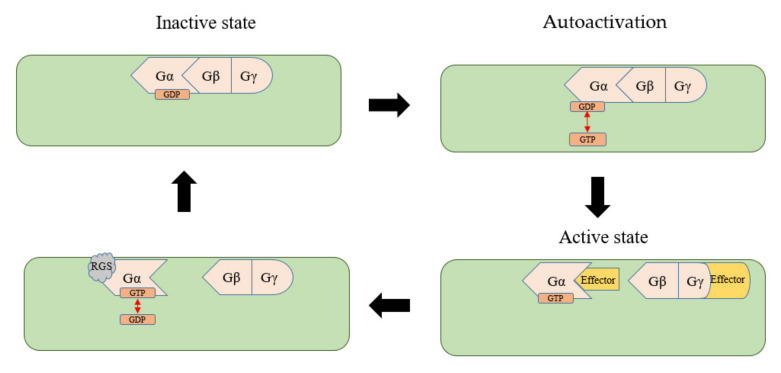
Traditional G-protein signaling cycle in plants.

**Figure 3 ijms-22-08678-f003:**
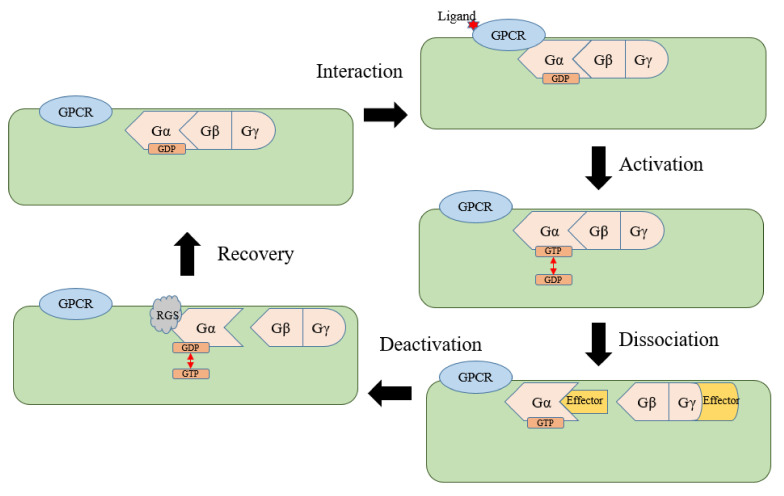
A proposed model of G-protein signaling cycle in plants.

**Table 1 ijms-22-08678-t001:** Response to plant hormones and characteristic morphological traits of Gα mutants.

Plant Hormone	*Arabidopsis gpa1*	Rice *rga1*	Reference
GA	Longer seed dormancy; hyposensitive to GA	Lower expression and activity of α-amylase induced by GA	[20,21,23]
BR	Hyposensitive to stomata closure, seed germination, hypocotyl, and root elongation induced by EBR	Weaken effect of root growth, and coleoptile and second coleoptile elongation stimulated by BR	[24,38]
ABA	Hyposensitive to ABA inhibition of stomatal opening; weaker activity of inward potassium channels; inhibition of elongation of primary roots	n. d.	[25,27]
MeJA	Hyposensitive to MeJA inhibition of stomatal opening; decreased ROS accumulation	n. d.	[28]
ET	Hyposensitive to ET promotion of stomatal closure	Hyposensitive to ET	[29,30]
Auxin	Reduced cell division in developing hypocotyls and leaves	n. d.	[31,32]
Phenotypes	Shorter Hypocotyl; less lateral root; weaker root avoidance of ATP	Shorter rod; round grain	[22,23,31,39,40]

**Table 2 ijms-22-08678-t002:** Putative GPCRs in plants.

Putative Plant GPCRs	Evidence of Interaction with Gα	Transmembrane Structure	Signal Pathways Involved	Reference
GCR1	Pull-down assays; yeast two-hybrid analysis; co-immunoprecipitation (Co-IP) assays	7-TM	Negative regulator of the GPA1-mediated ABA response; regulation of seed germination by GA and BR	[76,78]
GCR2	Bimolecular fluorescence complementation (BiFC); co-immunoprecipitation (Co-IP) assays	7-TM	ABA signal transduction	[79]
COLD1	Co-immunoprecipitation (Co-IP) assays; yeast two-hybrid analysis; bimolecular fluorescence complementation assays	9-TM	Confers chilling tolerance in japonica rice	[80]
CAND2	Bimolecular fluorescence complementation; yeast two-hybrid analysis	7-TM	Receptor of phytomelatonin	[82]

## Data Availability

Not applicable.

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
