# Peer review of "Research Advances in Heterotrimeric G-Protein α Subunits and Uncanonical G-Protein Coupled Receptors in Plants"

_ijms, 2021, doi:10.3390/ijms22168678_

Round 1
Reviewer 1 Report
The review manuscript from Liu et al. addresses an interesting and important topic: Are there GPCRs in plants and how does G protein signaling differ compared to animals. I think that such a review article would be needed by the scientific community.
However, a lot of phrases and explanations are misleading and therefore I cannot recommend publication in the current form. Starting in the abstract, G proteins and GPCRs are no second messengers (cAMP or IP1 are some examples here). Or '4. GPCR, a coupled Gα receptor'. Unfortunately the text contains a lot of these imprecise/wrong statements. Moreover, the term GPCR in its pure meaning, is a receptor that couples to G proteins. In that sense the authors are right to call those plant proteins GPCRs. However, this is misleading, because the term GPCR nowadays implies a certain protein architechture, activation mechanism and functionality. This is obviously different in plants and I would not recommend to use the same terms here.
While figure 1 not necessary, a figure showing a proposed structure of 'plant GPCRs' (e.g. GCR1, COLD1, ...) would be essential here.
While the topic is interesting, but the manuscript is in a premature state, I recommend a reconsideration of an updated version. I would suggest to let a GPCR pharmacologist read the manuscript before the next submission to avoid any misleading terms and phrases. Furthermore, I suggest the paragraph 5.1 to be extended (more details about the evidence for structure and function + a figure/table with the examples).
Author Response
Point 1: A lot of phrases and explanations are misleading and therefore I cannot recommend publication in the current form. Starting in the abstract, G proteins and GPCRs are no second messengers (cAMP or IP1 are some examples here). Or '4. GPCR, a coupled Gα receptor'. Unfortunately the text contains a lot of these imprecise/wrong statements.
Response 1: I have corrected all the errors and tried my best to present the views with precise descriptions. I have deleted the content of G-protein and GPCRs as a second messenger and corrected the description of GPCRs.
Point 2: Moreover, the term GPCR in its pure meaning, is a receptor that couples to G proteins. In that sense the authors are right to call those plant proteins GPCRs. However, this is misleading, because the term GPCR nowadays implies a certain protein architechture, activation mechanism and functionality. This is obviously different in plants and I would not recommend to use the same terms here.
Response 2: GPCR was first studied in animals. In animals, the term GPCR means a protein structure, activation mechanism and function. However, the completely different G protein activation mechanism in plants makes the definition of GPCR in animals not applicable in plants. Therefore, I have always described plant GPCR as a potential GPCR and gave it a new definition, that is, to avoid comparing it with GPCR in animals. Using the new definition of plant GPCR to study the potential GPCR can better understand the plant G-protein signaling pathway.
Point 3: While figure 1 not necessary, a figure showing a proposed structure of 'plant GPCRs' (e.g. GCR1, COLD1, ...) would be essential here.
Response 3: Figure 1 is a legend about the G-protein cycle in animals. Our article starts with the introduction of animal G-protein and then introduces it to plants, so I think Figure 1 needs to be shown. At the same time, we added the legend of traditional plant G-protein signal cycle, which can not only compare with animals, but also better introduce our new plant G protein cycle model. For proposed structure of plant GPCRs, because we define plant GPCRs as a multi-transmembrane protein rather than just 7 or 9 transmembrane protein, we do not limit it to a certain structure here and show their structure.
Point 4: Furthermore, I suggest the paragraph 5.1 to be extended (more details about the evidence for structure and function + a figure/table with the examples).
Response 4: In the paragraph 5.1, we not only added the potential plant GPCR TOM1, but also summarized the function, transmembrane structure and the evidence of interaction with Gα in Table 2. This can better explain their role in the G signaling pathway.
Reviewer 2 Report
This review is mainly focused on G-protein subunit Gα and GPCR receptors in plants. This review covers many research advances in the plants G-proteins but lacks some of the essential recent findings of plant G-proteins, especially about the Gα subunit. Several studies have explained the possibilities of the existence of plant GPCRs via functional research outcomes in that field. Comparing with the animal model, the authors have proposed a model. Authors should elaborate on the active role of plant GPCRs to support their model and include more references to make it pleasant.
Some more comments are added below,
- Title: Use G-Protein instead of G protein (“-” use standard form)
- Abstract
- G-Proteins and GPCR are not secondary messengers in signal transduction
- ‘Circulatory pathway’ wrong word in the statement
- “Both continue to ……..… to activate G proteins”: Effectors not activating G-proteins. “soybean RKN FR1” change to RK NFR1
- One of the references cited is a retracted article (Pandey et al., 2006 reference no.23). Please modify the text accordingly.
- Recent findings of GCR1 is missing in this review [(Eg. Chakraborty et al., 2019 (PMID: 30967583), Chakraborty et al., 2015 (PMID: 25668726)].
- The authors should explain the results of soybean G protein complementation in Arabidopsis.
- Many of the important recent findings of plant G-protein signalling are missing in this review [Eg. Maruta et al., 2019 (PMID: 31690635), Roy Choudhury et al. 2019 (PMID: 30622152), Kumar and Bisht, 2020 (PMID: 32875468), Jose et al. 2020 (PMID: 33011291), Guo et al. 2020 (PMID: 32847511), Sun et al. 2014 (PMID: 24777451).
- The style of representations of figures in this review is not correct. The heterotrimeric G-protein complex is a membrane anchored protein complex (Gα and Gγ). Similarly, the model of RGS is wrong as it is a membrane-bound protein.
Author Response
Point 1: Title: Use G-Protein instead of G protein (“-” use standard form)
Response 1: We have corrected all G proteins to G-protein in the whole review.
Point 2: Abstract.
Response 2: We revised some of the language and sentences of the abstract and described the contents with more accurate and appropriate words.
Point 3: G-Proteins and GPCR are not secondary messengers in signal transduction.
Response 3: We have deleted the content of G-protein and GPCRs as a second messenger.
Point 4: ‘Circulatory pathway’ wrong word in the statement.
Response 4: We have corrected ‘Circulatory pathway’ to ‘G-protein signal cycle’.
Point 5: “Both continue to ……..… to activate G proteins”: Effectors not activating G-proteins. “soybean RKN FR1” change to RK NFR1.
Response 5: We have described the effector as transmitting signal and we have corrected ‘soybean RKN FR1’ to ‘RK NFR1’.
Point 6: One of the references cited is a retracted article (Pandey et al., 2006 reference no.23). Please modify the text accordingly.
Response 6: For the retracted article, we deleted the relevant parts and modified the description.
Point 7: Recent findings of GCR1 is missing in this review [(Eg. Chakraborty et al., 2019 (PMID: 30967583), Chakraborty et al., 2015 (PMID: 25668726)].
Response 7: We have read and added these recent findings in this review.
Point 8: The authors should explain the results of soybean G protein complementation in Arabidopsis.
Response 8: Through soybean G protein complementation and increased content about experiment on two GPA1 mutants which have a single amino acid exchange, we come to the conclusion that plant G-proteins may also have mechanisms that function independently of nucleotide exchange. Therefore, this review will mainly focus on research in the canonical GDP-GTP exchange-dependent mechanism of G-protein.
Point 9: Many of the important recent findings of plant G-protein signalling are missing in this review [Eg. Maruta et al., 2019 (PMID: 31690635), Roy Choudhury et al. 2019 (PMID: 30622152), Kumar and Bisht, 2020 (PMID: 32875468), Jose et al. 2020 (PMID: 33011291), Guo et al. 2020 (PMID: 32847511), Sun et al. 2014 (PMID: 24777451).
Response 9: We have added the recent findings of plant G-protein signalling which you have mentioned.
Point 10: The style of representations of figures in this review is not correct. The heterotrimeric G-protein complex is a membrane anchored protein complex (Gα and Gγ). Similarly, the model of RGS is wrong as it is a membrane-bound protein.
Response 10: We modified the positional relationship between G protein and cell membrane, and modified RGS protein to a membrane-bound protein.
Round 2
Reviewer 1 Report
I was really surprised to get the updated version so fast. Basically because there were no big changes to the manuscript. While the topic is interesting and worth to be covered by a review article, I can support publication of this manuscript only with limited enthusiasm.
This is because of several reasons. First, I still think that mixing up terminology is misleading. Maybe animal GPCRs should be referred to as canonical GPCRs.
Second, I don't think this manuscript is timely. While the title states 'Research advances...', only 6 out of 75 references are from the last two years (2019-2021).
Third, paragraph 5 has been changed only incrementally. Information, I was looking for (and potential readers likely will) is how those plant GPCRs interact with the G proteins. What is known about structural requirements of this interaction and how do plant GPCRs interfere with GTP hydrolysis?
Comprehensively adressing these points is what the GPCR community is looking for. However, the authors make their points and suggest a 'new and unique' definition of plant GPCRs, which makes it a suitable (perspective/opinion) article, I don't want to hinder to get published. I would strongly suggest to let a GPCR parmacologist (canonical GPCRs) read the article in order to avoid misunderstandings by the overlapping terminology in the field.
Reviewer 2 Report
Authors have modified changes according to suggestions.
Author Response
We made some modifications in grammar and terminology.